# Trends in hospitalisation for urinary tract infection in adults aged 18–65 by sex in Spain: 2000 to 2015

Jesús Redondo-Sánchez[1,2,3☯], Ricardo Rodríguez-Barrientos[3,4,5]*, Mª del Canto de-Hoyos-Alonso[3,6], Cristina Muntañola-Valero[3,4,7], Isabel Almendro Martínez[8], Belén Peñalver-Argüeso[9], Carlos Fernández-Escobar[9], Ángel Gil-de Miguel[2], Isabel del Cura-González[2,3,4,5,10☯]

1 Ramon y Cajal Health Care Centre, Primary Care Management, Servicio Madrileño de Salud, Alcorcón, Madrid, Spain, 2 Department of Medical Specialities and Public Health, Rey Juan Carlos University, Alcorcón, Madrid, Spain, 3 Network for Research on Chronicity, Primary Care and Health Promotion (RICAPPS), Madrid, Spain, 4 Research Unit, Primary Care Management, Madrid Health Service, Madrid, Spain, 5 Instituto Investigación Sanitaria Gregorio Marañón IiSGM, Madrid, Spain, 6 Pedro Laín Entralgo Health Care Center, Primary Care Management, Madrid Health Service, Alcorcón, Madrid, Spain, 7 Fundación para la Investigación e Innovación Biosanitaria de Atención Primaria (FIIBAP), Madrid, Spain, 8 Facultativa del Servicio de Medicina Preventiva y Salud Pública, Hospital Universitario Gregorio Marañón, Madrid, España, 9 Unidad Docente de Medicina Preventiva y Salud Pública, Escuela Nacional de Sanidad— Instituto de Salud Carlos III, Madrid, Spain, 10 Karolinska Institutet and Stockholm University, Ageing Research Center, Stockholm, Sweden

☯ These authors contributed equally to this work.
* ricardo.rodriguez@salud.madrid.org

**Data Availability Statement:** The Ethics Committee of the Hospital Universitario Fundación Alcorcón has approved this research, including any

## Abstract

### Objective

To analyse trends in urinary tract infection (UTIs) hospitalisation among patients adults 18–65 aged in Spain from 2000–2015.

### Methods

Retrospective observational study using the Spanish Hospitalisation Minimum Data Set (CMBD), with codifications by the International Classification of Diseases (ICD-9). Variables: Type of UTIs (pyelonephritis, prostatitis, cystitis and non-specific-UTIs), sex, age (in 5 categories: 18–49 and 50–64 years in men, and 18–44, 45–55 and 56–64 years in women), comorbidity, length of stay, costs and mortality associated with admission. The incidence of hospitalisation was studied according to sex, age group and type of UTIs per 100,000. Trends were identified using Joinpoint regression.

### Results

From 2000–2015, we found 259,804 hospitalisations for UTIs (51.6% pyelonephritis, 7.5% prostatitis, 0.6% cystitis and 40.3% non-specific UTIs). Pyelonephritis predominated in women and non-specific UTIs in men. The hospital stay and the average cost (2,160 EUR (IQR 1,7872,540 were greater in men. Overall mortality (0.4%) was greater in non-specific UTIs. More women were admitted (rates of 79.4 to 81.7) than in men (30.2 to 41). The

potential data sharing. The CMBD data belong to the Ministry of Health and are partially accessible to the public. For data series on hospitalization, such as the data presented in this article, specific selections of anonymized microdata from CMBD records can be requested from the Ministry". The application form is available at: https://www.sanidad.gob.es/estadEstudios/estadisticas/cmbdhome.htm The data can be requested from the Research Unit of the Gerencia de Atención Primaria de la Comunidad de Madrid. uinvestigacion.ap@salud.madrid.org.

**Funding:** This study is funded by the Instituto de Salud Carlos III (ISCIII) through project PI19/01700 and RD21/0016/0027, co-funded by the European Union through funds from the European Recovery Instrument ("Next Generation EU"), in the framework of the Recovery, Transformation and Resilience Plan. The funders had no role in the study design, data collection and analysis, decision to publish, or preparation of the manuscript".

**Competing interests:** The authors have declared that no competing interests exist.

**Abbreviations:** AAPC, Average Annual Percent Change; APC, Annual Percent Change; CMBD, Spanish Hospitalization Minimum Data Set; ICD-9, International Classification of Diseases; IQR, Interquartile range; NHS, National Health Service; RECORD, Reporting of Studies Conducted using Observational Routinely-Collected Health Data; UTIs, Urinary tract infection.

greatest increase was found in men aged 50–64 years (from 59.3 to 87). In the Joinpoint analysis, the incidence of pyelonephritis increased in women [AAPC 2.5(CI 95% 1.6;3.4)], and non-specific UTIs decreased [AAPC −2.2(CI 95% −3.3;-1.2)]. Pyelonephritis decreased in men [AAPC −0.5 (CI 95% −1.5;0.5)] and non-specific UTIs increased [AAPC 2.3 (CI 95% 1.9;2.6)] and prostatitis increased [AAPC 2.6 (CI 95% 1.4;3.7)].

## Conclusions

The urinary infection-related hospitalisation rate in adults in Spain increased during the period 2000–2015. Pyelonephritis predominated in women and non-specific UTIs in men. The highest hospitalisation rates occurred in the women but the greatest increase was found in men aged 65–74. The lenght of stay and cost were higher in men.

## Introduction

Urinary tract infections [UTIs] are the most common bacterial infections in the community [1]. They affect men and women of all ages with an estimated incidence rate of 12.6% in women aged over 18 and 3% in men [2]. Except for a peak in women aged 14–24, prevalence increases with age [3]. 50–60% of women have at least one episode during their lifetime and up to 10% of postmenopausal women report having experienced an episode in the previous year [4].

In Spain, according to the latest study published by EPINE, of the total number of infections of community origin admitted to acute hospitals, the prevalence of urinary infections admitted is 19.9% [5], a similar figure, to that reported in other countries [17%] [6]. In recent years the rate of hospitalisation for UTIs has increased significantly according to some published studies [6–8]. This increase is more significant in men and older patients [6]. This has been associated with comorbidities such as incontinence, diabetes, neurogenic bladder and prostatic hyperplasia, polymedication and surgical instrumentation [6,9], situations that are more common as age increases.

Hospitalisation for UTIs has a major economic impact on health systems, carried significant mortality rates and high economic associated with treatment [1,10]. Economic expenditure is mainly related to an increase in length of stay per admission [11]. One of the determining factors for admission for UTIs is the increase in antimicrobial resistance, partly due to an inadequacy of empirical treatments, which should be adapted regularly to local antibiotic resistance models [9]. These resistances are related to isolated germs, which vary depending on the field studied, e.g. *Escherichia coli* appears in 74.2% of patients in the community, in 65.5% of hospitalised patients and in 46.6% of those in social health institutions [12].

Few studies describe hospital admissions for UTIs in patients in the age range of 18 to 64 years [13]. However, this type of studies are necessary to increase knowledge of the incidence rate for these infections in the community and in hospitals. They are essential for the planning and implementation of antimicrobial usage policies [14], bearing in mind that the factors determining hospital admission may differ depending on patient age range.

Our goal is to analyse the trend in hospital admissions for pyelonephritis, prostatitis, acute cystitis and unspecified urinary tract and non-specific urinary tract infection between the ages of 18 and 64 in Spain during the period 2000–2015.

## Methods

We conducted a retrospective observational study using the Spanish Hospitalisation Minimum Data Set (CMBD), a database provided by the Ministry of Health. The supporting RECORD checklist is available as supporting information; see S1 File.

### Participants and data source

Our study includes hospitalisations in adults (18–64 years) from 2000 to 2015, as during this period codifications remained homogeneous nationally, following the International Classification of Diseases, ninth version (ICD-9). Currently, around 99% of population in Spain is covered by the national health system. The CMBD is a register for all hospitals in Spain, which contains data on up to 97.7% of discharges from public hospitals during all periods, and from 2005 CMBD gradually started to include data from private hospitals, reaching a total coverage of 92% from all kinds of hospitals in 2014 [15]. The data were accessed for research purposes the 30 September 2020. We didn´t have access to information that could identify individual participants during or after data collection.

From the CMBD database we selected hospitalisations where the primary diagnosis had been a Urinary Tract Infection (UTIs). We considered UTIs as the diagnosis codes for pyelonephritis (590.10-11-3-80-81-9), acute (601.0) and chronic (601.1) prostatitis, cystitis (595.0-89-9), pregnancy-related UTIs (645.51-52-54-61-62-64; 646.53-60-63) and non-specific UTIs (599.0). Among hospitalisations with a primary diagnosis of non-specific UTIs, those who had a secondary diagnosis of pyelonephritis, acute or chronic prostatitis and cystitis were reclassified as such to maximise the chances of identifying specific UTIs diagnoses. Similarly, among hospitalisations with a primary diagnosis of pregnancy-related UTIs, those that had a secondary diagnosis of pyelonephritis and cystitis were reclassified. Finally, for the purposes of this study, we decided to combine pregnancy-related UTIs and non-specific UTIs into one group.

For each hospitalisation, we collected data on patient demographics (sex, age], type of discharge, dates of admission and discharge, primary and comorbid diagnosis, length of stay, and global cost. All the hospitalisations were grouped by age into categories: 18–49 years and 50–64 years in men, and 18–44 years, 45–55 years, and 56–64 years in women. We excluded readmissions, defined as admissions during the first 30 days after discharge in the same hospital.

### Statistical methods

Quantitative data are described with averages and standard deviation or medians with IQR (interquartile range), and categorical data are expressed with frequencies (absolute and percentages). Crude rates were defined as hospitalisations per 100,000 inhabitants aged 18–64, according to population data as at 1 July each year, provided by the Spanish National Institute of Statistics. This data was also used to obtain age-sex standardisation rates of both global and specific UTIs conditions that were calculated using 2015 Spanish population.

To identify possible varying trends over time, we performed a Joinpoint regression using age-adjusted UTIs rates, with the 2015 population as a reference. This widely-used approach fits a linear model when joinpoints (i.e., change points) are not known a priori and are to be estimated from the data. Two measurements are provided: An estimation of the annual percent change (APC) in each linear segment, and the average annual percent change (AAPC), computed as a weighted average of the APCs from the model. If no joinpoints were obtained, AAPC exactly reflects APC.

Statistical analysis was performed using Stata 14 and the Joinpoint Regression Program, version 4.5.0.1 (National Cancer Institute).

## Ethics statement

The study was approved by the Hospital U. Fundación Alcorcón Ethics Research Committee, (20/125) favourably evaluated by the Central Research Commission of the Primary Healthcare Management of Madrid (57/20). The need for consent was waived by the Ethics Committee (20/125).

## Results

During the period 2000–2015 there were a total of 259. 804 UTIs hospitalisations in adults aged 18 to 64 years, of which 134,119 were pyelonephritis (51.6%), 19,435 prostatitis (7.5%), 1,687 cystitis (0.6%), and 104,563 unspecified UTIs (40.3%) **Fig 1**, **Table 1**.

### Sociodemographic and clinical characteristics of admissions for UTIs

182,347 (70.2%) of the patients hospitalised were women. 42,432 of these women were pregnant (23.3% of women and 16.3% of all UTIs). In women, pyelonephritis admission predominated (61.9%) followed by unspecified UTIs (37.5%); in men, unspecified UTIs were 46.8%, PN 27.5%, and prostatitis 25.1%. Cystitis made up 0.7% of hospitalisations in women and 0.6% in men. In women, hospital admission was more common in the 18-44 age group (73.9%) for all types of UTIs (72.2% of pyelonephritis, 75.2% of cystitis and 71.9% of non-specific UTIs). In men, 55.3% of the hospitalisations were in the group aged 50 to 64 years. Hospital admission

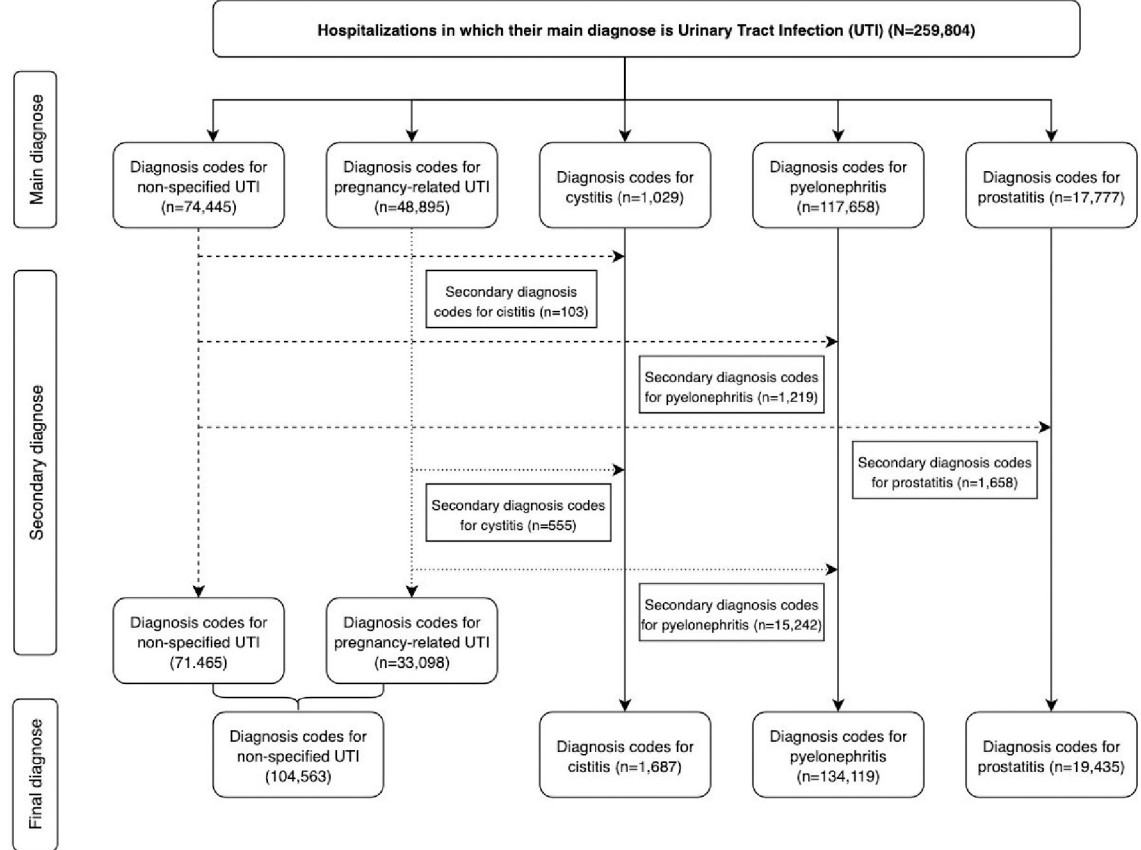

**Fig 1. Flow chart for the selection and classification of hospitalisations as per RECORD guidelines recommendations.**

**Table 1. Characteristics of hospital admissions for urinary tract infections.**

| | Total | Cystitis | | Pyelonephritis | | Prostatitis | Non-specific | |
|---|---|---|---|---|---|---|---|---|
| | | Males | Females | Males | Females | Males | Males | Females |
| | (n = 259.804) | (n = 492) | (n = 1195) | (n = 21303) | (n = 112816) | (n = 19435) | (n = 36227) | (n = 68336) |
| | n (%) | n (%) | n (%) | n (%) | n (%) | n (%) | n (%) | n (%) |
| **Age group—men (years), n (%)** | | | | | | | | |
| 18–49 | 34600 (44.7) | 176 (35.8) | - | 12091 (56.8) | - | 7666 (39.4) | 14667 (40.5) | - |
| ≥50 | 42857 (55.3) | 316 (64.2) | - | 9212 (43.2) | - | 11769 (60.6) | 21560 (59.5) | - |
| **Age group—women (years), n (%)** | | | | | | | | |
| 18–44 | 134786 (73.9) | - | 863 (72.2) | - | 84784 (75.2) | - | - | 49139 (71.9) |
| 45–55 | 26900 (14.8) | - | 177 (14.8) | - | 17350 (15.4) | - | - | 9373 (13.7) |
| 56–64 | 20661 (11.3) | - | 155 (13.0) | - | 10682 (9.5) | - | - | 9824 (14.4) |
| **Type of admission, n (%)** | | | | | | | | |
| Urgent | 251993 (97.0) | 292 (59.5) | 1073 (89.9) | 20565 (96.5) | 110295 (97.8) | 18925 (97.4) | 34590 (95.5) | 66253 (97.0) |
| Scheduled | 7752 (3.0) | 199 (40.5) | 121 (10.1) | 735 (3.5) | 2510 (2.2) | 506 (2.6) | 1624 (4.5) | 2057 (3.0) |
| **Previous conditions, n (%)** | | | | | | | | |
| Diabetes mellitus | 21992 (8.5) | 66 (13.4) | 72 (6.0) | 2430 (11.4) | 6513 (5.8) | 2112 (10.9) | 5634 (15.6) | 5165 (7.6) |
| Heart failure | 1000 (0.4) | 2 (0.4) | 3 (0.3) | 90 (0.4) | 255 (0.2) | 56 (0.3) | 280 (0.8) | 314 (0.5) |
| COPD | 3576 (1.4) | 21 (4.3) | 4 (0.3) | 671 (3.1) | 460 (0.4) | 589 (3.0) | 1445 (4.0) | 386 (0.6) |
| Renal failure | 3030 (1.2) | 9 (1.8) | 9 (0.8) | 338 (1.6) | 607 (0.5) | 133 (0.7) | 1105 (3.1) | 829 (1.2) |
| Liver disease | 2926 (1.1) | 6 (1.2) | 10 (0.8) | 298 (1.4) | 850 (0.8) | 190 (1.0) | 863 (2.4) | 709 (1.0) |
| Cancer | 13339 (5.1) | 40 (8.1) | 60 (5.0) | 1045 (4.9) | 3585 (3.2) | 496 (2.6) | 3868 (10.7) | 4245 (6.2) |
| Immunosuppression | 336 (0.1) | 2 (0.4) | 1 (0.1) | 16 (0.1) | 110 (0.1) | 13 (0.1) | 77 (0.2) | 117 (0.2) |
| HIV | 4074 (1.6) | 9 (1.8) | 8 (0.7) | 722 (3.4) | 1227 (1.1) | 239 (1.2) | 1021 (2.8) | 848 (1.2) |
| Muscle paralysis | 2092 (0.8) | 4 (0.8) | 2 (0.2) | 212 (1.0) | 90 (0.1) | 31 (0.2) | 1354 (3.7) | 399 (0.6) |
| Pregnancy | 42432 (16.3) | - | 500 (41.8) | - | 15524 (13.8) | - | - | 26408 (38.6) |
| **Cost (€), (median, IQR)** | 2160 (1787–2540) | 2598 (2015–3105) | 1900 (1304–2398) | 2201 (1900–2974) | 2064 (1795–2463) | 2176 (1687–2256) | 2579 (2017–3257) | 1992 (1345–2540) |
| **Length of stay (days), (median, IQR)** | 4 (2–6) | 3 (2–6) | 3 (1–4) | 4 (3–7) | 4 (2–6) | 3 (2–5) | 5 (3–7) | 3 (2–6) |
| **Length of stay (days), n (%)** | | | | | | | | |
| 0–3 | 116992 (45.0) | 251 (51.0) | 791 (66.2) | 8414 (39.5) | 49349 (43.7) | 9895 (50.9) | 13062 (36.1) | 35230 (51.6) |
| 4–7 | 104660 (40.3) | 144 (29.3) | 293 (24.5) | 8886 (41.7) | 49597 (44.0) | 7897 (40.6) | 14798 (40.8) | 23045 (33.7) |
| 8–11 | 23426 (9.0) | 37 (7.5) | 62 (5.2) | 2352 (11.0) | 9260 (8.2) | 1197 (6.2) | 4653 (12.8) | 5865 (8.6) |
| 12–15 | 7620 (2.9) | 24 (4.9) | 19 (1.6) | 775 (3.6) | 2610 (2.3) | 257 (1.3) | 1780 (4.9) | 2155 (3.2) |
| >15 | 7106 (2.7) | 36 (7.3) | 30 (2.5) | 876 (4.1) | 2000 (1.8) | 189 (1.0) | 1934 (5.3) | 2041 (3.0) |
| **Type of discharge, n (%)** | | | | | | | | |
| Home | 252116 (97.0) | 481 (97.8) | 1168 (97.7) | 20576 (96.6) | 109999 (97.5) | 19116 (98.4) | 34620 (95.6) | 66156 (96.8) |
| Transfer/Others/Unknown | 6673 (2.6) | 9 (1.8) | 22 (1.8) | 633 (3.0) | 2694 (2.4) | 309 (1.6) | 1223 (3.4) | 1783 (2.6) |
| **Mortality rate, n (%)** | 1015 (0.4) | 2 (0.4) | 5 (0.4) | 94 (0.4) | 123 (0.1) | 10 (0.1) | 384 (1.1) | 397 (0.6) |

for pyelonephritis were in the youngest men aged 18 to 49 years (56.8%); for the remaining UTIs, admission was more common in the 50–64 age group (cystitis 64.2%, prostatitis 60.6% and non-specific UTIs 59.5%). Regarding the comorbidities studied, the most frequent pathology was diabetes (8.5%), followed by cancer (5.1%). The mean length of stay was 4 days, one day longer in the case of men admitted for a non-specific UTIs. 85.3% remained hospitalised for less than one week. Extended stays, of twelve days or more, were found in 5.6% of cases and were more common in men, for all types of UTIs. The median cost of hospitalisation was

**Table 2. Urinary tract infections hospital admissions rates per 100,000 inhabitants by sex and age group, Spain, 2000–2015.**

| Year/Age year | Women | | | | Men | | |
|---|---|---|---|---|---|---|---|
| | Total | 18–44 | 45–55 | 56–64 | Total | 18–49 | 50–64 |
| 2000 | 79.4 | 95.3 | 45.0 | 56.2 | 30.2 | 21.7 | 59.3 |
| 2001 | 80.8 | 97.3 | 44.1 | 58.8 | 30.2 | 21.0 | 59.6 |
| 2002 | 80.3 | 97.2 | 46.2 | 53.3 | 29.1 | 19.7 | 59.3 |
| 2003 | 82.4 | 100.2 | 46.2 | 55.7 | 30.5 | 20.0 | 64.1 |
| 2004 | 82.0 | 99.0 | 48.7 | 56.2 | 30.0 | 19.3 | 63.8 |
| 2005 | 79.6 | 96.3 | 45.9 | 57.3 | 30.6 | 19.3 | 66.0 |
| 2006 | 82.5 | 100.8 | 46.6 | 58.1 | 32.4 | 19.9 | 71.1 |
| 2007 | 80.6 | 97.9 | 48.5 | 56.3 | 31.8 | 19.8 | 68.5 |
| 2008 | 82.1 | 98.8 | 52.2 | 58.1 | 31.5 | 19.4 | 68.1 |
| 2009 | 77.0 | 93.0 | 48.2 | 55.9 | 31.9 | 18.7 | 70.5 |
| 2010 | 75.1 | 88.7 | 50.5 | 59.5 | 32.6 | 19.0 | 71.2 |
| 2011 | 73.2 | 85.8 | 50.6 | 59.7 | 34.9 | 19.3 | 77.4 |
| 2012 | 74.1 | 87.8 | 51.5 | 58.7 | 34.1 | 18.7 | 74.8 |
| 2013 | 78.5 | 92.0 | 58.0 | 62.2 | 37.9 | 21.1 | 80.5 |
| 2014 | 80.6 | 95.1 | 58.9 | 64.9 | 39.3 | 21.2 | 83.2 |
| 2015 | 81.7 | 95.3 | 60.5 | 69.9 | 41.0 | 21.3 | 87.0 |

€2,160 (IQR 1,787–2,540), higher for men in all types of UTIs. Overall mortality was 0.4%, higher in unspecified UTIs in both men and women at 1.1% and 0.6% respectively. Table 1 details patient characteristics overall and for each of the different infection types by sex. (Table 1)

## Hospitalisation rates by age and type of UTIs

The rate of hospitalisation for urinary tract infection for the period 2000–2015, standardised by age and sex, varied in women within a range of 79.4 per 100,000 inhabitants in 2000 to 81.7 per 100,000 inhabitants in 2015. In men these rate were lower, although they increased progressively over time, ranging from 30.2 per 100,000 inhabitants in 2000 to 41.0 per 100,000 in 2015 (Table 2). Under the age analysis, hospitalisation rates among younger patients remained stable, with higher figures in women aged 18–44 (between 85.5 and 100.8 per 100,000) than in men aged 18–49 (between 18.7 and 21.7 per 100,000). However, in the older age groups, admission rates increased in both women aged 45 to 64 and, in particular, in men aged 50 to 64, with rates rising from 59.3 per 100,000 in 2000 to 87 in 2015.

By UTIs type, the highest hospitalisation rates in women were due to pyelonephritis, which saw a progressive increase over the years with a minimum from 40.8 in 2000 to 55.5 in 2015. Behind this were admissions with a diagnosis of non-specific UTIs, which dropped from 37.7 (2000) to 25.8 (2015). In men, the highest admission rates were for non-specific UTIs which, unlike the pattern seen in women, increased from a low of 12.5 (2000) to a high of 19.5 in 2015. Prostatitis also increased progressively, falling within a range of 7 to 11.7 per 100,000 inhabitants, while cases of pyelonephritis in men remained more stable (rates of 8.3–10 per 100,000). S2 File details the rates for each of the different infections by sex and age groups.

## Trend analysis

The overall trend analysis with the Joinpoint regression model showed a significant increase for the period with an AAPC of 0.8 (CI 95% 0.1 to 1.5), with two inflection points, one in 2007

and another in 2011, identifying an early upward trend in the rate of hospitalisations, rising from 2000 to 2007, with an annual percentage of change (APC) of 0.5 (CI 95% −0.2 to 1.2), followed by a downward trend from 2007 to 2011, with APC of −1.5 (CI 95% −4.0 to 1.0), and another upturn from 2011 to 2015, this time statistically significant, with an APC of 3.8 (CI 95% 2.1 to 5.4).

Different results were found upon analysing these results by sex, adjusting for age in each group. In men, the AAPC between 2000 and 2015 was 1.2 (CI 0.8 to 1.6), with a single inflection point in 2010, which gave rise to two trends, both upward. For women, the AAPC between 2000 and 2015 was 0.4 (CI 0.0 to 0.8), and two turning points were obtained in 2008 and 2011, resulting in three different trends, the first trend was ascendant from 2000 to 2008, the next one was descendent from 2008 to 2011, and the last one ascendant from 2011 to 2015 (Fig 2).

When stratifying these results by sex, considering the different types of UTIs, clear differences between women and men were observed for pyelonephritis and non-specific UTIs. (Fig 3). Pyelonephritis increased in women: the AAPC between 2000 and 2015 was 2.5 (95% CI 1.6 to 3.4), with two significant upward trends, from 2000–2006 and from 2011–2015. In men, cases of pyelonephritis decreased in this period with an AAPC-0.5 (CI 95% −1.5 to 0.5). Non-specific UTIs decreased in women with AAPC −2.2 (CI 95% −3.3 to −1.2) and two marked trends; there was a steady increase in men throughout the period with AAPC 2.3 (CI 95% 1.9 to 2.6). Cases of prostatitis increased from 2000 to 2015 with AAPC 2.6 (CI 95% 1.4 to 3.7) with two trends and significant increase from 2008 to 2015. Fig 3 shows the trends in hospital admission rates by type of ITUs and by sex. Although on this graph cases of cystitis appear to be stable as hospitalisation rates are very low throughout the period compared to the other types of infection, in women the hospitalization trend was descendent with an the AAPC −5.1 (CI 95% −7.0 to −3.2) between 2000 and 2015 and in men the tendency was ascending since 2003 with APC 2.9, CI 95% (0.1 to 5.8). S3 File contained the graphs for cystitis with an axis range that allows these differences to be seen.

Additional file 3. Trends in hospitalization rates due for cystitis in S3 File.

The trend analysis in UTI hospital admission rates by age group and by type of UTIs is shown in Fig 4 for women and Fig 5 for men.

**Pyelonephritis in women.** In women, hospitalisation rates for pyelonephritis increased in all age groups, more markedly in the 18 to 44 age group, with a statistically significant increase in AAPC between 2000 and 2015 of 2.8 (95% CI 1.9 to 3.7.) In this group, 3 trends were identified, two of them ascending and statistically significant, from 2000 to 2006. In women aged 45–55 years there is an increase in the AAPC period of 2.2 (CI 95% 1.1 to 3.3) with two upward trends with a point of inflection in 2011. In women aged 56 to 64, there was an insignificant global increase from 2000 to 2015 and two contradicting trends, with an inflection point in 2012 and then a significant upward trend, between 20012–2015 APC 6.3 (95% CI 0.2 to 12.7) (Fig 4).

**Non-specific UTIs in women.** In women, non-specific UTIs behaved differently by age group. In 18–44 age group, non-specific UTIs dropped AAPC −4.2 (CI 95% −6.5 to −1.9) with two inflection points in 2008 and 2011 and a significant downturn from 2000 to 2008. In the 45–55 age group there is a significant increase from 2000 to 2015 with AAPC/APC at 1.9 (CI 95% 1.4 to 2.5). In the 56–64 age group there is a constant upward trend in the study period APPC/APC 2.1 CI95% (1.5 to 2.7) (Fig 4).

**Cystitis in women.** Significant decrease in women aged 18 to 44 years. AAPC −7.5 (CI 95% −10 to −5). No significant differences in the rest of the age groups (Fig 4)

**Pyelonephritis in men.** In the 18–49 age group there was a significant increase at AAPC/APC 0,9 (0.3 to 1.5); in men aged 50 to 64 there is an insignificant increase of AAPC/APC 0.1

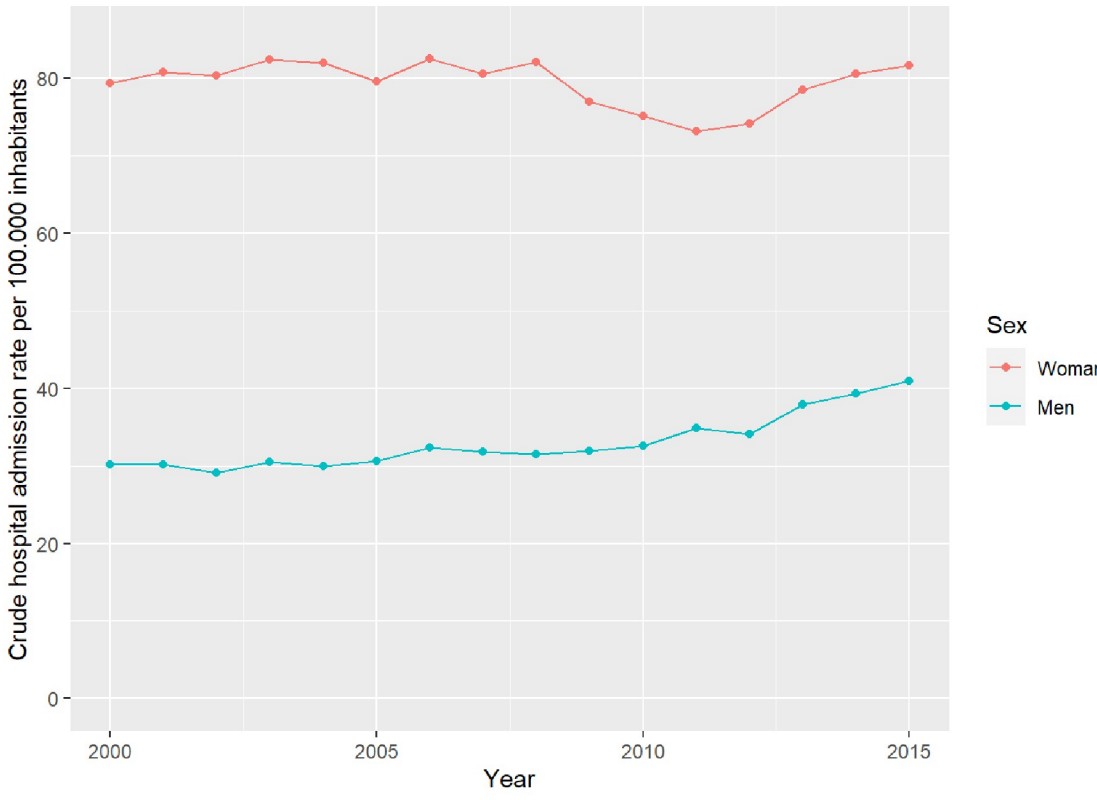

| | Women | Men |
|---|---|---|
| **APPC** | **0,4** | **1.2** |
| **(CI 95%)** | (0.0 – 0.8) | (0.8 – 1.6) |
| **APC** | **0.4** | **0.8** |
| **(CI 95%)** | (-0.1 to 1.0) | (0.2 to 1.4) |
| **Years** | 2000-2008 | 2000-2010 |
| | **-3.2** | **3.3** |
| | (-7.7 to -1.6) | (1.7 to 5) |
| | 2008-2011 | 2010-2015 |
| | **3.7** | |
| | (2.1 to 5.2) | |
| | 2011-2015 | |

APC= annual percent change, AAPC average annual percent change

**Fig 2. Trends in urinary tract infections hospital admissions in the population aged 18–64 years, Spain, 2000–2015.**

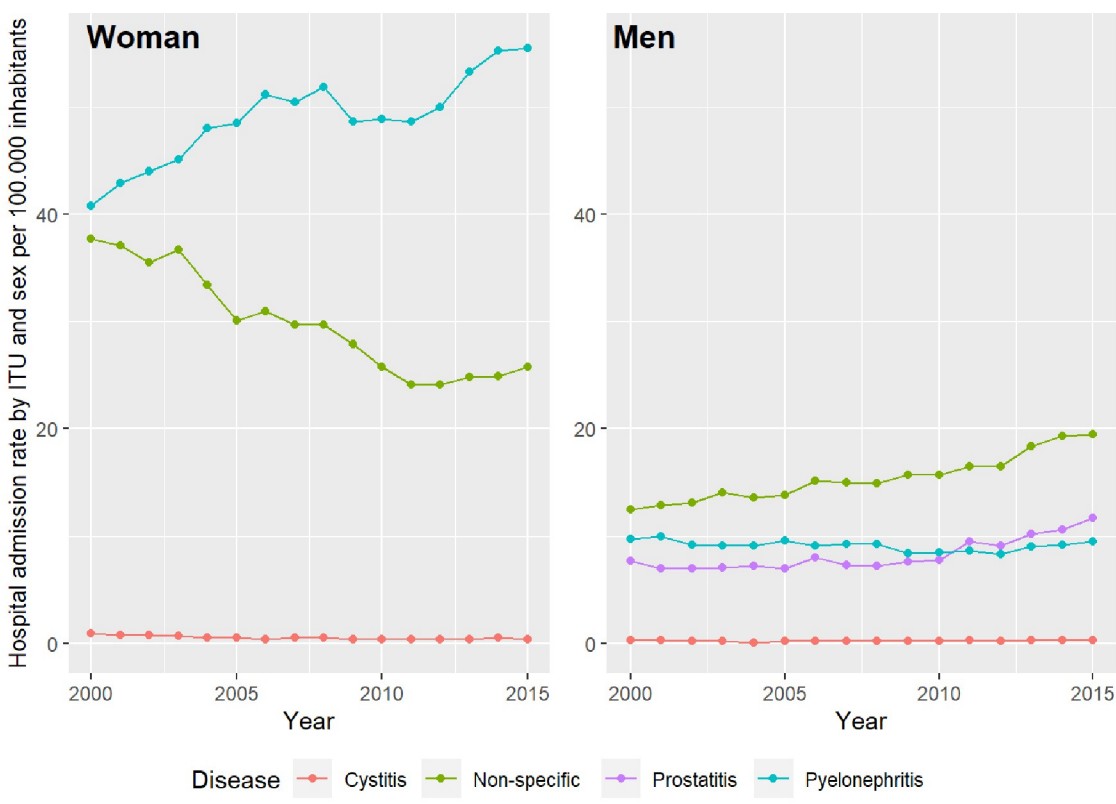

| | Women | | | Men | | | |
|---|---|---|---|---|---|---|---|
| | Pyelonephitis | Cystitis | Non-specified | Pyelonephitis | Cystitis | Non-specified | Prostatytis |
| **APPC (CI 95%)** | 2,5 (1,6 to 3,4) | -5.1 (-7 to-3.2) | -2.2 (-3.3 to -1.2) | -0.5 (-1.5 to 0.5) | -1.7 (-6.1 to 3.1) | 2.3 (1.9 to 2.6) | 2.6 (1.4 to 3.7) |
| | 3.8 (2.5 to 5.1) 2000-2006 | -11.02 (-14.7 to -7.6) 2000-2006 | -3.3 (-3.8 to -2.7) 2000-2012 | -1.3 (-1.9 to -0.7) 2000-2012 | -17.9 (-35.3 to 4.2) 2000-2003 | 2.3 (1.9 to 2.6) 2000-2015 | 0.2 (-1.5 to 2.0) 2000-2008 |
| **APC (CI 95%) Years** | -0,6 (-2,7 to 1,5) 2006-2011 | -0.9 (-3.4 to 1.7) 2006-2015 | 1.9 (-3.4 to 7.6) 2012-2015 | 2.9 (-2.2 to 8.2) 2012-2015 | 2.9 (0.1 to 5.8) 2003 -20015 | | 5.3 (3.4 to 7.2) 2008-2015 |
| | 4,4 (2,3 to 6,6) 2011-2015 | | | | | | |

APC= annual percent change, AAPC average annual percent change

**Fig 3. Trends in hospital admission rates by type of urinary tract infections and by sex.**

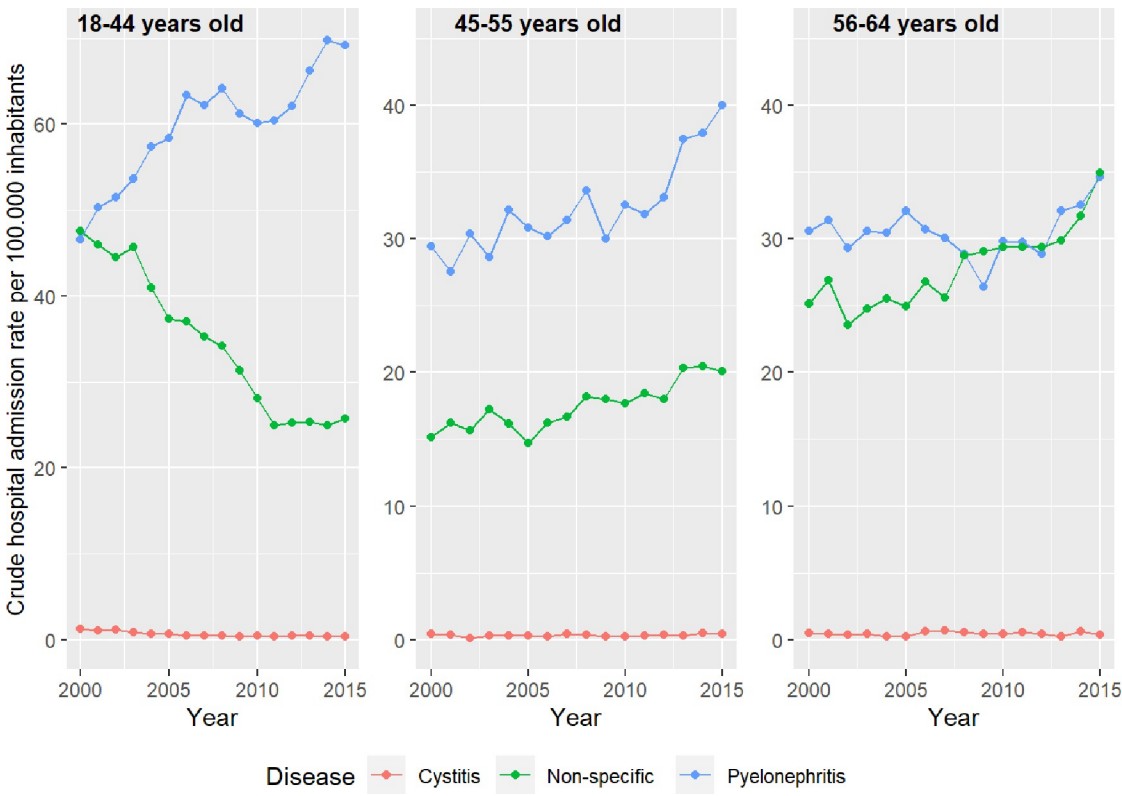

**Fig 4. Trends in urinary tract infections hospital admission rates by age group and by type of UTIs in women.**

(IC 95% −0.3 to 0.5) for the whole period, with neither age group showing changes in trend (Fig 5).

**Non-specific UTIs in Men.** Significant increase from 2000 to 2015 in both age groups at AAPC/APC 0.9 (CI 95% 0.3 to 1.5) in the 18–49 group y AAPC/APC 3 CI 95% (2.6 to 3.5) in the 50–64 age group (Fig 5).

**Prostatitis.** There is an increase in both age groups, show two trends, tipping in 2008 and with a significant increase from 2008 to 2015. Over this period, for the 18–49 age group, the APC was 4.3 (CI 95% 1.5 to 7.1) while in the 50–64 age group, the APC was 5.8 (CI 95% 4.0 to 7.6) (Fig 5).

**Cystitis in men.** Insignificant overall decline in both age groups, with two trends in the 18–49 age group and a significant upward trend from 2002 to 2015 with APC 5.8 (CI 95% 0.5 to 11.4) (Fig 5).

## Discussion

Our study found that hospitalisations for UTIs in Spain increased between 2000 and 2015 overall and in both sexes, following a similar pattern to the one found in other studies [6,7,13,16]. The sharpest increase in hospital admission took place in the past 4–5 years. Hospitalisations were highest in women aged 18 to 44, but during the study period the greatest increase was seen in men aged 50 to 64. In women, admissions for pyelonephritis predominated. In men, it was hospitalisations for non-specific UTIs. Admissions for cystitis were low in both sexes.

259,804 admissions for UTIs were analysed for the period 2000–2015. Of these, 42,432 were pregnant women (23% of total women and 16% of total UTI admissions). 70% of UTIs hospitalisations were in women, with an admission rate twice that of men.

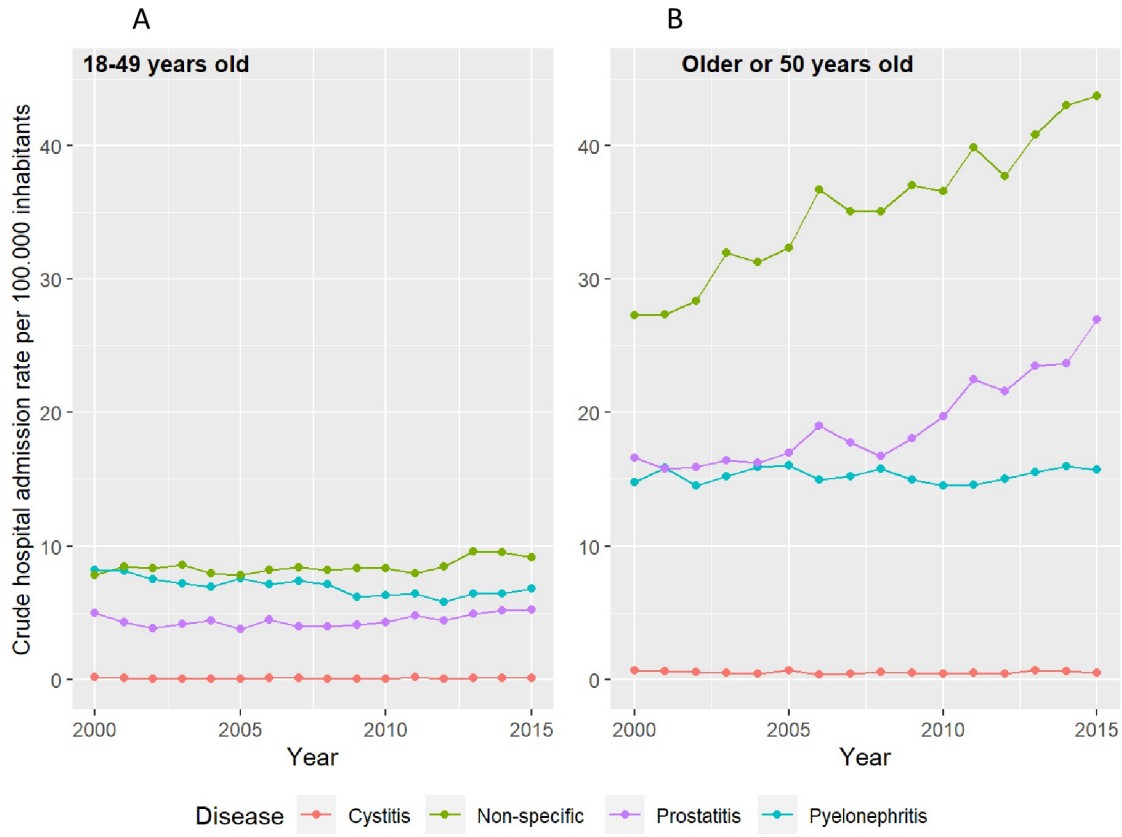

| | 18-49 years | | | | 50-64 years | | | |
|---|---|---|---|---|---|---|---|---|
| | Pyelonephitis | Cystitis | Non-specified | Prostatytis | Pyelonephitis | Cystitis | Non-specified | Prostatytis |
| **APPC (CI 95%)** | 0.9 (0.3 to 1.5) | -2.6 (-11.7 to 7.5) | 0.9 (0.3 to 1.5) | 1.1 (-0.4 to 2.7) | 0.1 (-0.3 to 0.5) | -0.4 (-2.4 to 1.7) | 3.0 (2.6 to 3.5) | 3.4 (2.2 to 4.5) |
| **APC (CI 95%), Years** | 0.9 (0.3 to 1.5) 2000-2015 | -43.0 (-73.2 to 21.5) 2000-2012 | 0.9 (0.3 to 1.5) 2000-2015 | -1.5 (-3.8 to 0.8) 2000-2008 | 0.1 (-0.3 to 0.5) 2000-2015 | -0.4 (-2.4 to 1.7) 2000-2015 | 3.0 (2.6 to 3.5) 2000-2015 | 1.3 (-0.5 to 3.1) 2000-2008 |
| | | 5.8 (0.5 to 11.4) 2012-2015 | | 4.3 (1.5 to 7.1) 2008-2015 | | | | 5.8 (4.0 to 7.6) 2008-2015 |

APC= annual percent change, AAPC average annual percent change

**Fig 5. Trends in urinary tract infections hospital admission rates by aged group and by type of UTIs in men.**

In terms of age, admission patterns in adult UTIs was inverse in men and women. This means that the greatest number of admissions were in younger women and older men. In women, the 18–44 age group had hospitalisation incidence rates ranging from 92% to 101%, almost double those recorded for women aged 45–55 years and 56–64 years. In the case of

men, admissions were highest in the 50–64 age group, 2–4 times those of men aged 18 to 49 years. This group of men aged 50–64 was where there was the greatest increase in admission rate (from 59 to 87 per 100,000). The higher incidence of UTIs admissions in younger women may be related to pregnancy. In men aged 50–64 years, the higher rate of admission may be due to the fact that, from this age, there is an increase in the presence of obstructive prostate pathology, urolithiasis and complex urinary infections, and an increased use of urological instrumentation [17].

Regarding types of UTIs, there were also different patterns depending on sex. Pyelonephritis increased in women and decreased in men, while in the case of non-specific UTIs the opposite happened (they reduced in women and increased in men). Hospitalisations for cystitis were very low in both sexes although it decreases in young women but increases in young men. One explanation for this difference may be poorer antibiotic management in males of this age in terms of choice of antibiotic type, dose and duration.

Acute pyelonephritis is epidemiologically very similar to urinary tract infections in general, with a higher incidence in women than in men and peaks during childbearing age [18]. In our study, pyelonephritis is the most frequent diagnosis in women, with an increasing incidence during the study period (41 per 100,000 in 2000 and 55.5 in 2015), well above the admission rates for men (between 8 and 10%). Our data are similar to those found other studies in terms of age and sex [13,18,19]. In a classical study [19] on acute pyelonephritis this difference between sexes is reported, with 87% of cases in women and rates of admission for pyelonephritis in women double those for men.

75% of pyelonephritis in women occurred between the age of 18–44, a group where the trend of increase was greater. It should be noted that during pregnancy there may be a higher incidence of pyelonephritis and UTIs [13] with a risk of increased preterm births. The higher number of admissions in women aged 15 to 44 in our study, in addition to pregnancy, could be related to sexual behavior (sexual intercourse in the previous 30 days, a new sexual partner in the previous year or the recent use of spermicide), given that these has been related to a larger number of acute pyelonephritis [20]. Diabetes and urinary incontinence also independently increase the risk of pyelonephritis [20]. Although both factors are more commonly associated with peri- or postmenopausal women, a high proportion of young women with incontinence prior to the episode of pyelonephritis has been found (20), which would merit further investigation.

In our study, prostatitis accounted for 7.5% of the total number of admissions, and 25% of admissions for UTIs in men. They were more frequent in the 50–64 age group (61%). The admission rate for prostatitis increased from 8% in 2000 to 12% in 2015, with a trend increasing from 2008 onwards, mainly in men aged 50–64. In cases of prostatitis a dual mode pattern was identified, with an initial peak of incidence at 20–30 years and a second peak from 60 onwards [21,22]. The risk factors associated with prostatitis are a previous history, prostate hypertrophy or cancer, and urological interventions such as prostate biopsies [23,24]. Transrectal prostate biopsy has led to a progressive increase in hospitalisations for secondary bacterial prostatitis, often difficult to manage due to the progressive increase in resistance rates to fluoroquinolones [23]. These risk factors explain why prostatitis is more common in the older age group. Other studies have associated the increase in hospitalisation rate in people over 50 years has also been linked to a history of urinary tract interventions, neoplasms and antimicrobial prescriptions in the three months prior to admission [25].

Non-specific UTIs showed similar patterns in men to prostatitis, increasing admission rates from 12.5 in 2000 to 19.5 per 100,000 in 2015 with a greater increase in the 50–64 age group. In women, non-specific UTIs decreased overall from 38% to 26% due to the downturn in the 18–44 age group. However, the admission trend for this type of UTIs increases in older

women. Other studies have found that this increase with age persists in both sexes from the age of 65, where non-specific UTIs were the most prevalent infections and, in addition, the ones that increased the most, from 1.5 to 3.6 per 1000 (150 to 360 per 100,000) in habitants from 2000 to 2015 (8). In or study, as well as others [6], non-specific UTIs are associated with greater mortality. Given that they are also associated with greater comorbidity, and in older men it can be assumed that, like in the over-65s, these may be due to catheter infections or other complex urinary tract infections [6], and further studies are necessary to characterise them more precisely.

Comorbidities were associate more with non-specific UTIs and were more common in men for all different types of UTI. The most frequent associated comorbidity was diabetes, with 8.5% of admissions for UTIs, followed by oncological processes (5%). Diabetes is a risk factor for UTIs in both men and women and increases the chance of hospitalisation. Studies show that diabetics are two to four times more at risk of being hospitalised with urinary tract infections than non-diabetics [26]. This risk increases over time [26]. In other studies, 15% of admissions for UTIs aged under 65 were in diabetic patients [13], and this proportion increases in patients aged over 65 from 21–32% [8,13].

The average stay was 4 days: 85% of the cases were admitted for a maximum of 7 days, although in 3% of cases this extended to 15 days or more. The mean length of stay was higher in pyelonephritis and non-specific UTIs and lower in less common infections (cystitis and prostatitis). Lengthier stays were more common in men, which may influence their higher cost per stay compared to women. Although not all studies analyse the same population and age groups, the length of stay was similar to that published for UTIs hospitalisations in the USA (4–5 days) [7,16], and lower than those in Korea and Japan, with average stays of 8 days in patients admitted for pyelonephritis [13,18]. The average length of stay is an indirect measure of the quality of UTIs management, in that appropriate use of empirical treatment and antimicrobial agents reduces the length of stay and associated costs [17], and resistance to three antimicrobials is responsible for longer duration of admission and higher cost of these processes [16].

In our study the percentage of deaths among patients admitted for UTIs was 0.4%. It was higher in both sexes for non-specific UTIs (1.1% in men and 0.6% in women) than for other UTIs. In pyelonephritis, mortality in men was higher (0.4% vs 0.1% in women). In another study we performed on this same database, mortality in people over 65 years of age in Spain for this period was 5.4%, 6% in non-specific UTIs and 2.4% in pyelonephritis [8]. Our mortality data appear lower than in other studies, although the comparison is complicated since the same types of UTIs are not necessarily analysed and it is difficult to find a breakdown of mortality for different age groups. In one Japanese study, mortality among patients admitted for pyelonephritis in patients under 65 was 1%, compared to 5.6% in patients over 65 years of age [13], which is higher than our pyelonephritis data for under 65s and similar for those aged over 65. With regard to sex, a Korean study also found higher mortality from pyelonephritis in men (rates of 1.5 in men and 0.5 in women per 10.000 cases, considering all age groups] with an overall mortality of 0.6 [27]. In other studies on complicated UTIs, mortality was higher in those admitted with catheter-related infections than non-catheter related ones (3.4% vs 2.8%, across all ages] [6], which could partly explain the higher mortality rate for non-specific UTIs. In addition to the type of UTIs, mortality has been associated with age, men, comorbidity, and the use of drugs such as anti-diabetics, corticosteroids, or immunosuppressants [13].

Limitations of this study include those related to the use of a database, CMBD, which contains administrative data of hospital discharges in Spain registered by the doctors themselves, so there may be biases in relation to the registration and its interpretation. In addition, this database does not include data related to the organisms causing the different types of UTIs or

the associated antibiotic treatment. It cannot be ruled out that the same person had been admitted twice or more during the study period, (except for admissions within 30 days of first admission, which were considered readmissions and excluded from this study], but from an epidemiological point of view, we believe that this possible bias does not drastically affect the data.

The results obtained in this study provide a necessary assessment of the changes occurring over 16 years in the incidence of hospitalization for UTIs, with important data from an epidemiological point of view. These data are necessary for the correct contextualization of urinary tract infections and their connection with the implementation of good practices in the use of antimicrobials, as well as improving certain clinical procedures, to reduce the number of hospital admissions and have a favorable impact on patients and the health system as a whole.

## Conclusions

Hospitalizations for UTIs in adults in Spain increased during the period 2000–2015 with some different patterns depending on patient sex. Pyelonephritis predominated in women and non-specific UTIs in men. There were more admissions in women, but it was in men where the admission trend increased the most (in the 65-74-year-old group] and where the length of stay and cost were higher. The UTIs associated with the highest mortality were non-specific UTIs.

## Supporting information

**S1 File. The RECORD statement–checklist of items, extended from the STROBE statement, that should be reported in observational studies using routinely collected health data.** Title: Trends in hospitalisation for urinary tract infection in adults aged 18–65 by sex in Spain: 2000 to 2015.
(DOCX)

**S2 File. Type of urinary tract infections hospital admissions rates per 100,000 inhabitants by sex, Spain, 2000–2015.**
(DOCX)

**S3 File. Trends in hospitalizations rates due to cystitis.**
(DOCX)

## Acknowledgments

To our colleagues from the Research Unit for their support: Marcial Caboblanco-Muñoz, Juan Carlos Gil-Moreno y Elena Polentinos-Castro

## Author Contributions

**Conceptualization:** Jesús Redondo-Sánchez, Ricardo Rodríguez-Barrientos, Isabel del Cura-González.

**Data curation:** Jesús Redondo-Sánchez, Ricardo Rodríguez-Barrientos, Mª del Canto de-Hoyos-Alonso, Cristina Muntañola-Valero, Isabel Almendro Martínez, Belén Peñalver-Argüeso, Carlos Fernández-Escobar, Isabel del Cura-González.

**Formal analysis:** Jesús Redondo-Sánchez, Ricardo Rodríguez-Barrientos, Mª del Canto de-Hoyos-Alonso, Cristina Muntañola-Valero, Isabel Almendro Martínez, Belén Peñalver-Argüeso, Carlos Fernández-Escobar, Isabel del Cura-González.

**Funding acquisition:** Jesús Redondo-Sánchez, Ricardo Rodríguez-Barrientos, Isabel del Cura-González.

**Investigation:** Jesús Redondo-Sánchez, Ricardo Rodríguez-Barrientos, Isabel del Cura-González.

**Methodology:** Jesús Redondo-Sánchez, Ricardo Rodríguez-Barrientos, Isabel del Cura-González.

**Project administration:** Jesús Redondo-Sánchez, Ricardo Rodríguez-Barrientos, Ángel Gil-de Miguel, Isabel del Cura-González.

**Resources:** Jesús Redondo-Sánchez, Isabel del Cura-González.

**Software:** Jesús Redondo-Sánchez, Isabel del Cura-González.

**Supervision:** Jesús Redondo-Sánchez, Ricardo Rodríguez-Barrientos, Mª del Canto de-Hoyos-Alonso, Isabel del Cura-González.

**Validation:** Jesús Redondo-Sánchez, Isabel del Cura-González.

**Visualization:** Jesús Redondo-Sánchez, Isabel del Cura-González.

**Writing – original draft:** Jesús Redondo-Sánchez, Mª del Canto de-Hoyos-Alonso, Isabel del Cura-González.

**Writing – review & editing:** Jesús Redondo-Sánchez, Ricardo Rodríguez-Barrientos, Isabel Almendro Martínez, Belén Peñalver-Argüeso, Carlos Fernández-Escobar, Ángel Gil-de Miguel, Isabel del Cura-González.

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
