## [Decision Letter · Decision Letter 0]

6 Dec 2023

PONE-D-23-35805"Trends in hospitalization for urinary tract infection in adults population 18-65 aged by sex in Spain: 2000 to 2015"PLOS ONE

Dear Dr. Rodríguez-Barrientos,

Thank you for submitting your manuscript to PLOS ONE. After careful consideration, we feel that it has merit but does not fully meet PLOS ONE’s publication criteria as it currently stands. Therefore, we invite you to submit a revised version of the manuscript that addresses the points raised during the review process.

We look forward to receiving your revised manuscript.

Kind regards,

Kwame Kumi Asare, Ph.D

Academic Editor

PLOS ONE

Journal Requirements:

2. Thank you for stating the following financial disclosure: "This study is funded by the Instituto de Salud Carlos III (ISCIII) through project PI19/01700 and co-funded by the European Union.".

5. Please amend either the title on the online submission form (via Edit Submission) or the title in the manuscript so that they are identical.

Reviewers' comments:

Reviewer's Responses to Questions

**Comments to the Author**

1. Is the manuscript technically sound, and do the data support the conclusions?

Reviewer #1: Yes

Reviewer #2: Yes

2. Has the statistical analysis been performed appropriately and rigorously? 

Reviewer #1: Yes

Reviewer #2: Yes

3. Have the authors made all data underlying the findings in their manuscript fully available?

Reviewer #1: Yes

Reviewer #2: Yes

4. Is the manuscript presented in an intelligible fashion and written in standard English?

Reviewer #1: Yes

Reviewer #2: Yes

5. Review Comments to the Author

Reviewer #1: First of all, I deeply appreciate the opportunity to review this article. I find it an utmost important issue in population health.

As it is mentioned in the 3rd paragraph starting from the end, survival bias might be taken into consideration when considering retrospective cohorts collected through real-word datasets, as populations with reiterative admissions (on average, proobably milder) might be overrepresented. This might be specially important when considering mortality and admissions rates. In the article, it is stated that "from a strictly epidemiological point of view we do not believe that this is relevant.". Could you further explain me why you do consider it is not relevant?

Within results, it is stated: "there is actually a downward turn for cystitis in both sexes". Could you further explain about this and whether you do think this significant negative AAPC finding in women might be attibutable to both sexes?

Within methods, it is explained that the population coverage has increased over your period. In your article, you explain brilliantly about significant trends, and about the detection of reflection points. Rather than the trends themselves, do you believe your results could rule out the explanation of an overall UTI increasing trend due to an increasing population coverage during your period of study?

About figure 2:

- Please review pyelonephritis in men, where line after unique reflection point has a 2.9 APC and 95% IC is -2.2 to 88.2.

- Please consider English translation of "hombres" and "mujeres"

About figure 3:

- Please review non-specified UTI in 56-64 years. It is not clear whether the period from 2003 to 2004 is another line (stable?).

- Please review pyelonephritis APC in 18-44 years is 5.0 and a 95% CI of 3.7 to 4.0.

- Please review the two overlapped periods in pyelonephritis in 56-64 years (2002-2012, 2000-2015).

About figure 4:

- Please review cystitits APC from 2000 to 2002 in the 18-49 years group, and whether confidence intervals do not seem to be centered.

- Please review cystitits in the group of 50-64 years AAPC, and whether .7 would mean "0.7" in this text.

Thank you again for this interesting research article.

Regards.

Reviewer #2: Review

Introduction - Unfortunately the entire premise of the study is fraught with issue given how many admissions for UTIs are in actuality due to likely other comorbidities with incidentally found ASB. The authors at least in part need to discuss this because publishing data that suggests the commonality of UTIS requiring admission will increase concern on the part of providers and patients/families.

one way to potentially get around this is frame the whole paper that you looked at UTI diagnosis upon admission - i know that's semantics but we can't know if they actually had a UTI or if they gave the person an inappropriate diagnosis.

Methods -

Was there rationale for age groupings? Is this just how it is reported in database?

Methods are somewhat limited but i suppose are appropriate for what they are trying to accomplish

typo at beginning of results ITU - not UTI

i am surprised by this number - it feels like it would be higher over a 15 year period. It would be helpful to know the denominator (what percentage of all admissions are for this reason)

I would not repeat all of the information from table 1 in the body of results. pick one or the other

overall the results are just dense and hard to follow - the figures are helpful and maybe when the paper is published that will flow better but i would try and pare down some of the wordier sections

Discussion

i think the question for many of these groups is not necessarily why they are getting UTIs but why are they getting admitted for UTIs....even pyelonephritis shouldn't really require an admission in an otherwise healthy individual

i do think the very low mortality should be highlighted and how men had higher length of stay and higher cost (which you do do)

6. PLOS authors have the option to publish the peer review history of their article (what does this mean?). If published, this will include your full peer review and any attached files.

Reviewer #1: **Yes: **Francisco Rodríguez Cabrera

Reviewer #2: No

---

## [Author Response · Author response to Decision Letter 0]

23 Jan 2024

Dear Editor,

Thank you very much for your interest in our manuscript and for all the points you have made, which undoubtedly improve the proposal.

Following your indications, the PLos ONE style requirements have been revised in naming the files. Full information on the funding of the study has been included. Thank you very much for including the modification to the funding. The correct information is as follows: 

"This study is funded by the Instituto de Salud Carlos III (ISCIII) through project PI19/01700 and RD21/0016/0027, co-funded by the European Union through funds from the European Recovery Instrument ("Next Generation EU"), in the framework of the Recovery, Transformation and Resilience Plan. The funders had no role in the study design, data collection and analysis, decision to publish, or preparation of the manuscript".

In relation to availability of the data, as stated in our first letter, the data from this study are only available upon request, because there are legal or ethical restrictions on sharing data publicly. The data contain potentially confidential information and are owned by an external organisation, in this case the Ministry of Health. To obtain the data one must follow an application procedure at the Ministry, indicating the periods and diagnoses of interest, and have the approval of an ethics committee.

We include the following information, which was validated by PLoS ONE in the publication of another article that used the same data source in a population over 65 years of age. Redondo-Sánchez J, et al. PLoS One. 2021 Sep 29;16(9):e0257546. doi: 10.1371/journal.pone.0257546. PMID: 34587191; PMCID: PMC8480842.

"The Ethics Committee of the Hospital Universitario Fundación Alcorcón has approved this research, including any potential data sharing. The CMBD data belong to the Ministry of Health and are partially accessible to the public. For data series on hospitalization, such as the data presented in this article, specific selections of anonymized microdata from CMBD records can be requested from the Ministry". The application form is available at: https://www.sanidad.gob.es/estadEstudios/estadisticas/cmbdhome.htm

The title in the online submission form (via Edit Submission) has been modified to match the title of the manuscript, which was checked to ensure it was identical in both sections. The file titles have been included in the "Supporting Information files" section and the citations have been checked in the text to confirm that they match.

In accordance with the journal's guidelines for exceeding 20,000 characters, the response to the reviewers is provided in the attachment Respond review.pdf 

We remain at your disposal for any further clarification. 

Best Regards

Ricardo Rodríguez  

PONE-D-23-35805

"Trends in hospitalization for urinary tract infection in adults population 18-65 aged by sex in Spain: 2000 to 2015".

PLOS ONE

Reviewer #1: First of all, I deeply appreciate the opportunity to review this article. I find it an utmost important issue in population health.

As it is mentioned in the 3rd paragraph starting from the end, survival bias might be taken into consideration when considering retrospective cohorts collected through real-word datasets, as populations with reiterative admissions (on average, probably milder) might be overrepresented. This might be especially important when considering mortality and admissions rates. In the article, it is stated that "from a strictly epidemiological point of view we do not believe that this is relevant.". Could you further explain me why you do consider it is not relevant?

Thank you very much for your positive consideration of our work. We agree with the reviewer that the way it has been written is perhaps very assertive. The arguments on which we based this statement are, firstly, that the data are reported annually, so patients who had more than one episode would only be over-represented in that time period and, secondly, that the number of readmissions is very low compared to the number of single episodes. Taking into account the reviewer's consideration, we have modified the wording of the sentence in the article to read "but from an epidemiological point of view, we believe that this possible bias does not drastically affect the data". Discussion Section, p. 13, line 415.

Within results, it is stated: "there is actually a downward turn for cystitis in both sexes". Could you further explain about this and whether you do think this significant negative AAPC finding in women might be attributable to both sexes?

As expected from a clinical point of view, the number of admissions for cystitis is very low compared to the rest of the UTI categories. This explains why their trend is not very visible, appearing as flat curves when plotted together with the others. For this reason, supplement 4 had been added to make the differences in this type of UTI visible. 

The joint-point analysis shows that, during the study period, there is a downward trend in females over the whole period and in males only until 2003, after which it increases significantly (APC 2.9, 95%CI (0.1 to 5.8)). This increase is due to young males, as shown in figure 5, and could be related to poorer antibiotic management in this age group in terms of choice of antibiotic type, dose and duration, while not taking into account that "cystitis" in males is a complicated UTI.

A brief clarification has been included in the results and discussion section, taking into consideration the reviewer's suggestions:

Results section, page 8, lines 239-242 reads as follows:

"Although on this graph cases of cystitis appear to be stable as hospitalisation rates are very low throughout the period compared to the other types of infection, in women, hospitalisation for cystitis fell with the AAPC -5.1 (CI 95% -7.0 to -3.2) between 2000 and 2015 and in men it tended to rise from 2003 with APC 2.9, CI 95% (0.1 to 5.8)".

Discussion section, page 11, lines 323-326 reads as follows:

"Hospitalisations for cystitis were very low in both sexes although it decreases in young women but increases in young men. One explanation for this difference may be poorer antibiotic management in males of this age in terms of choice of antibiotic type, dose and duration.

Within methods, it is explained that the population coverage has increased over your period. In your article, you explain brilliantly about significant trends, and about the detection of reflection points. Rather than the trends themselves, do you believe your results could rule out the explanation of an overall UTI increasing trend due to an increasing population coverage during your period of study?

During the study period, the Spanish population increased but the percentage population coverage of our health system remained similar. Spain's National Health System is publicly funded, providing universal health care free of charge at the point of use. The incidence of UTI is related to the population at each time point and the rates are adjusted for age and sex. Considering the results for the different types of UTI, one sees that incidence is related to age and the presence of unspecified UTI, so that the most likely cause of the rising trend in these pathologies may be the associated comorbidity and associated instrumentation (catheterisation, etc.).

About figure 2: 

- Please review pyelonephritis in men, where line after unique reflection point has a 2.9 APC and 95% CI is -2.2 to 88.2. - 

Please consider English translation of "hombres" and "mujeres".

Thank you very much for the indication. There was an error in the transcription of the data which has been corrected. The correct figure is not 88.2 but 8.2. The suggested translation correction has been included in the figure (now figure 3).

About figure 3:

- Please review non-specified UTI in 56-64 years. It is not clear whether the period from 2003 to 2004 is another line (stable?).

(Now figure 4).

There is a constant trend from 2000 to 2015 in the case of non-specified UTI in the 56-64 age group. The trend remained constant throughout the study period. In this case the APC and APPC are equal, as the APPC is a summary measure of the APCs. This nuance has been included in the text to make it clearer. The text reads as follows: 

"In the 56-64 age group there is a constant upward trend in the study period APPC/APC 2.1 CI95% (1.5 to 2.7) (Figure 4).

- Please review pyelonephritis APC in 18-44 years is 5.0 and a 95% CI of 3.7 to 4.0.

The indicated APC interval correction has been included. The lower and upper 95% CI values are 3.7 and 6.4. 

 - Please review the two overlapped periods in pyelonephritis in 56-64 years (2002-2012, 2000-2015).

The correction has been included. The correct periods are 2002-2012 and 2012-2015. 

About figure 4: 

- Please review cystitis APC from 2000 to 2002 in the 18-49 years group, and whether confidence intervals do not seem to be centred.

(Now figure 5).

The cystitis APC value from 2000 to 2012 in the 18-49 years group is -43 (-73.2 to 21.5). 

There was an error in the first annual period of change, which we have corrected, so that it is not from 2000-2002, but from 2000-2012.

- Please review cystitis in the group of 50-64 years AAPC, and whether .7 would mean "0.7" in this text.

The value has been revised and corrected. The APC range is 95%CI -2.4 to 1.7.

Reviewer #2: Review

Introduction - Unfortunately the entire premise of the study is fraught with issue given how many admissions for UTIs are in actuality due to likely other comorbidities with incidentally found ASB. The authors at least in part need to discuss this because publishing data that suggests the commonality of UTIS requiring admission will increase concern on the part of providers and patients/families.

one way to potentially get around this is frame the whole paper that you looked at UTI diagnosis upon admission - i know that's semantics but we can't know if they actually had a UTI or if they gave the person an inappropriate diagnosis.

Thank you very much for your considerations, which have undoubtedly helped us to improve our results. As the reviewer indicates, having a UTI does not exclude other diagnoses on admission or other reasons for admission, and many patients requiring admission for very different pathologies may have an accompanying urinary tract infection. In our study, the data source used, Minimum Basic Data Set (MBDS), includes a principal diagnosis and the rest of the diagnoses as secondary. In this study, the inclusion criterion was UTI as the main diagnosis on admission. Following the reviewer's recommendation, this semantic consideration has been qualified in the text. 

Methods - 

Was there rationale for age groupings? Is this just how it is reported in database? 

Methods are somewhat limited but i suppose are appropriate for what they are trying to accomplish

The age groupings were considered in the project design phase in the project application to the funding body. This took into account studies carried out in other contexts using similar age ranges and the expected changes in incidences by sex in the different age groups among the adult population. 

In relation to the methods proposed, those chosen are appropriate to meet the objectives.

To calculate the rates, a standardisation of rates by age and sex was carried out. Crude rates of mortality, morbidity and other health events are one of the summary measures of the experience of each population that facilitate this comparative analysis. However, comparison of crude rates may be inappropriate, particularly when population structures are not comparable in terms of factors such as age, sex, socio-economic status or any others that determine the magnitude of crude rates and may distort their interpretation. The calculation of specific rates, determined in well-defined subgroups, is a way to avoid certain confounding factors. Adjusted rates allow for more valid comparisons between populations.

Kramer S. Clinical Epidemiology and Biostatistics. A primer for Clinical Investigators and Decision-makers. Berlin Heidelberg, German: Springer-Verlag. 1988

Szklo M, Nieto J. Epidemiology, Beyond the basics. Gaithersburg, MD: Aspen Publishers, Inc. 2000.

Xunta de Galicia, Consellería de Sanidade e Servicios Sociais. Organización Panamericana de la Salud, Programa Especial de Análisis de Salud. Análisis Epidemiológico de Datos Tabulados (Epidat), Versión 4.2 [29 de julio de 2016). 

Joinpoint regression models10 were used for trend analysis using the software provided by the US National Cancer Institute Surveillance Research Program. The purpose of these models is twofold: to identify the time at which significant changes in the trend occur and to estimate the magnitude of the observed increase or decrease in each interval. In this way, the years (period) that make up each trend were expressed in the results, as well as the annual percentage change (APC) and the confidence intervals for each trend. Standardised hospitalisation rates were used to estimate these models.

This method has been widely used in trend studies.

Patel K, Hamedani AG, Taneja K, Koneru M, Wolfe J, Sprankle K, Patel P,Mullen MT, Siegler JE. Differential thrombectomy utilization across hospitalclassifications in the United States. J Stroke Cerebrovasc Dis. 2023 Dec;32(12):107401. doi: 10.1016/j.jstrokecerebrovasdis.2023.107401. Epub 2023 Oct 27. PMID: 37897885.

Wang X, Dong B, Huang F, Zhang J, He R, Du S, Zhang J, Ma J, Wang H, Zhang B,Liang W. Temporal Trends in Cardiovascular Health Status Among Chinese School-Aged Children From 1989 to 2018: Multiwave Cross-Sectional Analysis. JMIR PublicHealth Surveill. 2023 Oct 23;9:e45564. doi: 10.2196/45564. PMID: 37870895;PMCID: PMC10628687.

Cayuela A, Cayuela L, Escudero-Martínez I, Rodríguez-Domínguez S, González A, Moniche F, Jiménez MD, Montaner J. Analysis of cerebrovascular mortality trends in Spain from 1980 to 2011. Neurologia. 2016 Jul-Aug;31(6):370-8. English, Spanish. doi: 10.1016/j.nrl.2014.09.002. Epub 2014 Dec 15. PMID: 25524042.

Orozco-Beltran D, Cooper RS, Gil-Guillen V, Bertomeu-Martinez V, Pita-Fernandez S, Durazo-Arvizu R, Carratala-Munuera C, Cea-Calvo L, Bertomeu-Gonzalez V, Seoane-Pillado T, Rosado LE. Trends in mortality from myocardial infarction. A comparative study between Spain and the United States: 1990-2006. Rev Esp Cardiol (Engl Ed). 2012 Dec;65(12):1079-85. English, Spanish. doi: 10.1016/j.recesp.2012.02.026. Epub 2012 Jun 22. PMID: 22727798.

Márquez-Calderón S, Pérez Velasco L, Viciana-Fernández F, Fernández Merino JC. Tendencia de la mortalidad por edad y sexo en España (1981-2016). Cambios asociados a la crisis económica [Trends in age-sex-specific mortality in Spain (1981-2016). Changes associated with the economic crisis]. Gac Sanit. 2020 May-Jun;34(3):230-237. Spanish. doi: 10.1016/j.gaceta.2019.03.007. Epub 2019 Jun 4. PMID: 31174896.

typo at beginning of results ITU - not UTI

The typographical error has been corrected. "259. 804 UTIs hospitalisations".

i am surprised by this number - it feels like it would be higher over a 15 year period. It would be helpful to know the denominator (what percentage of all admissions are for this reason)

Our data source is the CMBD until 2015 and the RAE-CMBD, which was implemented in 2016 as a new data model for the Minimum Basic Data Set for Hospital Discharges, extending the register to areas other than hospitalisation (day hospital, highly complex technical and procedural departments and emergency departments) and to the private sector. 

This register provides access to microdata on hospitalisations by the pathology requested. It does not provide the total number of admissions for any pathology. What it does issue at present are reports with global information. The last two public reports that are available correspond to the years 2020 and 2019. https://www.sanidad.gob.es/estadEstudios/estadisticas/docs/RAE-CMBD_Informe_Hospitalizacion_2019.pdf

In 2020, the most frequent diseases according to the main diagnosis, taking into account the ICD-10-ES classification, were diseases of the respiratory system which, adding the cases of the special chapter of emerging codes, established for COVID-19 infection, totalled 16.86% of admissions, followed in frequency by diseases of the circulatory system (13.31%), diseases of the digestive system (11.51%) and neoplasms (10.04%). However, given the exceptional nature of admissions in 2020 due to the COVID-19 pandemic, we decided to review the general public information offered by the registry for 2019, which allows us to orientate the magnitude, but not to use this information as possible denominators for the calculation suggested by the reviewer. 

The Specialised Healthcare Activity Register (RAE-CMBD) included a total of 16,641,436 contacts for the year 2019. A total of 89.76% (14,936,847) took place in the National Health System (NHS) and 10.24% in private centres. Of the total number of contacts, 4,549,270 corresponded to inpatient care. Of these, 83.14% took place in NHS hospitals. The activity rate was 802 contacts per 10,000 inhabitants. Most of the contacts recorded in hospitalisation corresponded to acute care hospitals, representing 99.10% of the total. More than half of the hospital contacts in the NHS corresponded to women (51.71%). By age group, the highest number of discharges was in the 75 years and over age group (31.29%).

By chapters of the International Classification of Diseases (ICD-10-ES), diseases of the circulatory system, with 13.66% of all discharges, were the most frequent cause of hospitalisation, followed by diseases of the respiratory system (12.98%), diseases of the digestive system (12.15%) and neoplasms (10.04%). These four chapters account for almost one in every two discharges (48.83%).

By diagnostic group, according to the clinical classification system CCS (HCUP Project Classification System), the most frequent were pneumonia (except those caused by tuberculosis or STD), chronic obstructive pulmonary disease and other diseases of the lower respiratory tract. The 15 most frequent diagnoses in medical cases accounted for 38% of discharges. As can be seen, urinary tract infections independently were the fourth most frequent reason for admission by diagnosis. 

I would not repeat all of the information from table 1 in the body of results. pick one or the other

 overall the results are just dense and hard to follow - the figures are helpful and maybe when the paper is published that will flow better but i would try and pare down some of the wordier sections

Following the reviewer's indications, both the contents of the text related to table 1 and the results section have been reduced. We propose putting the previous supplement 3 as figure 2 in order so the results can be seen in the article itself.

Discussion 

I think the question for many of these groups is not necessarily why they are getting UTIs but why are they getting admitted for UTIs ....even pyelonephritis shouldn't really require an admission in an otherwise healthy individual

We agree with your assessment of the interest of studying both the indications for admission in the different processes and the variability in the application of these criteria in different health centres and in different contexts, rural and urban areas, public and private centres, insurance coverage, etc. Although the information available in our study does not allow us to address this question, it is undoubtedly an interesting line of research.

I do think the very low mortality should be highlighted and how men had higher length of stay and higher cost (which you do do)

We agree with your comment. The difference in mortality and mean length of stay between sexes and types of UTI is a data point that we find of great interest and as such it has been reflected in the results and in the discussion.

---

## [Decision Letter · Decision Letter 1]

2 Feb 2024

TRENDS IN HOSPITALISATION FOR URINARY TRACT INFECTION IN ADULTS AGED 18-65 BY SEX IN SPAIN: 2000 TO 2015.

PONE-D-23-35805R1

Dear Dr. Rodriguez-Barrientos,

We’re pleased to inform you that your manuscript has been judged scientifically suitable for publication and will be formally accepted for publication once it meets all outstanding technical requirements.

Kind regards,

Kwame Kumi Asare, Ph.D

Academic Editor

PLOS ONE

Additional Editor Comments (optional):

Reviewers' comments:

Reviewer's Responses to Questions

**Comments to the Author**

1. If the authors have adequately addressed your comments raised in a previous round of review and you feel that this manuscript is now acceptable for publication, you may indicate that here to bypass the “Comments to the Author” section, enter your conflict of interest statement in the “Confidential to Editor” section, and submit your "Accept" recommendation.

Reviewer #1: All comments have been addressed

Reviewer #2: All comments have been addressed

2. Is the manuscript technically sound, and do the data support the conclusions?

Reviewer #1: Yes

Reviewer #2: Yes

3. Has the statistical analysis been performed appropriately and rigorously? 

Reviewer #1: Yes

Reviewer #2: Yes

4. Have the authors made all data underlying the findings in their manuscript fully available?

Reviewer #1: Yes

Reviewer #2: Yes

5. Is the manuscript presented in an intelligible fashion and written in standard English?

Reviewer #1: Yes

Reviewer #2: Yes

6. Review Comments to the Author

Reviewer #1: The authors of the scientific article have meticulously addressed several critical aspects that I think underscore the manuscript's credibility and validity. Firstly, the manuscript's technical soundness is evident through the clear alignment between the study goals, the data presented and the conclusions drawn. The statistical analysis, a cornerstone of the manuscript's integrity, has been executed with appropriate rigor and sophistication, employing suitable methodologies that enhance the reliability of the findings. Lastly, the manuscript's presentation is coherent and accessible, written in standard English that ensures the scientific community can easily understand and engage with the content. This combination of technical soundness, rigorous statistical analysis and clear communication effectively addresses key questions of scientific rigor, making the manuscript a valuable contribution to its field.

Reviewer #2: Thank you for responding to concerns. I feel that the paper is now acceptable for publication if the other reviewers agree.

7. PLOS authors have the option to publish the peer review history of their article (what does this mean?). If published, this will include your full peer review and any attached files.

Reviewer #1: **Yes: **Fran Rodríguez Cabrera

Reviewer #2: No

---

## [Editor Report · Acceptance letter]

3 Apr 2024

PONE-D-23-35805R1 

PLOS ONE

Dear Dr. Rodríguez-Barrientos, 

I'm pleased to inform you that your manuscript has been deemed suitable for publication in PLOS ONE. Congratulations! Your manuscript is now being handed over to our production team.

Kind regards, 

on behalf of

Dr. Kwame Kumi Asare 

Academic Editor

PLOS ONE